# Increased Soluble CMG2 Serum Protein Concentration Is Associated with the Progression of Prostate Carcinoma

**DOI:** 10.3390/cancers11081059

**Published:** 2019-07-26

**Authors:** Thomas Greither, Marios Marcou, Paolo Fornara, Hermann M. Behre

**Affiliations:** 1Center for Reproductive Medicine and Andrology, Martin Luther University Halle-Wittenberg, Ernst-Grube-Str. 40, 06120 Halle (Saale), Germany; 2Clinics for Urology, Martin Luther University Halle-Wittenberg, Ernst-Grube-Str. 40, 06120 Halle (Saale), Germany

**Keywords:** prostate cancer, prognosis, serum biomarker, CMG2

## Abstract

Prostate carcinoma (PCa) is one of the leading causes of cancer-related death in males, but biomarkers for the prognosis are rare. Capillary morphogenesis gene 2 (CMG2) is a modulator of extracellular matrix remodeling during angiogenesis. Four isoforms of CMG2 have been described so far, one secreted in the serum as soluble CMG2 (sCMG2). The aim of this study was to evaluate the sCMG2 serum concentrations in 179 PCa patients and 163 age-matched control subjects by ELISA and correlate it to clinical and demographic parameters. We observed that sCMG2 concentration is increased in the serum of PCa patients with metastases, while no significant differences in the concentrations were detected between the control subjects and patients with localized PCa. Furthermore, elevated sCMG2 concentrations were significantly associated with the highest T stage. Increased sCMG2 serum concentrations tended to be associated with a worsened overall and disease-specific survival of the PCa patients. In conclusion, sCMG2 may be an interesting additive biomarker for the prediction of the progression of PCa and the patients’ outcome.

## 1. Introduction

Prostate carcinoma (PCa) is still the most frequent tumor entity in males in many countries [1,2], and the second most common cancer entity worldwide [3]. Mainly owing to its overall incidence, it is also the second to third highest cause of cancer-related death of males in developed countries [1,3]. Although being a relatively slow growing tumor entity, which is frequently localized, metastasization remains an urgent clinical problem. Metastasization of a prostate carcinoma is considered a sign for increased aggressiveness of the tumor, and thus is associated with an increased risk of tumor-related death [4]. Owing to the lack of adequate prognostic biomarkers, personalized management of PCa remains difficult [5,6]. Actually, prostate-specific antigen (PSA) is used for diagnosis of PCa from liquid biopsies; however, still bearing several limitations [7,8]. Other biomarkers for diagnosis and prognosis of PCa, like PCA3 [9,10,11], miR-375 [12,13], urokinase-typ plasminogen activator receptor (uPAR) [14,15], or combinations of biomarkers [16,17], are controversially discussed in the literature, with limited impact on the actual clinical management of PCa so far. Therefore, the evaluation of biomarkers from liquid biopsies with accurate diagnostic or prognostic impact remains an important task in translational PCa research.

Capillary morphogenesis gene 2 (CMG2) is a 489 amino acid (aa) transmembrane protein, which was identified by its involvement in the development of 3D capillary structures in vitro [18]. It is also one of the receptors enabling the anthrax toxin uptake in the cell, therefore, alternatively named anthrax toxin receptor 2 (ANTXR2) [19]. Under normal physiological conditions, CMG2 is discussed as a potent regulator of the extracellular matrix (ECM) by regulation of matrix metalloproteinase (MMP) activity [20], thereby supporting processes like angiogenesis and endothelial cell proliferation [20,21,22]. Although the pathways regulated by CMG2 are not fully elucidated yet, high affinity binding partners like laminin and collagen IV are described as well as target genes of CMG2 like MMP2 [23] or collagen VI [24]. CMG2 exists in various isoforms, including a 386 aa (isoform 2) and 322 aa (isoform 3), besides the 489 aa full length variant [25], with isoform 3 lacking the transmembrane domain, and thus is suggested to be secreted [18,19]. Soluble CMG2 (sCMG2) has been investigated mainly in its role as anthrax toxin inhibitor thus far [19,26,27], although potentially also interesting in the tumor-induced remodeling of the ECM and induction of angiogenesis. 

The aim of this study was to measure sCMG2 concentration in the serum of PCa patients and age-matched controls, and to correlate the data with clinical and prognostic parameters of the PCa patients.

## 2. Results

### 2.1. sCMG2 Concentration in the Serum of Control Subjects and PCa Patients

We measured the concentration of sCMG2 in the serum of 179 PCa patients and 163 age-matched control subjects. Additionally, we analyzed sCMG2 concentration in the serum of 55 patients with other cancer entities (for demographic and clinical details of all subjects, see Table 1). Of the 179 PCa patients, 114 patients exhibited localized tumors, while metastases were found in 65 patients (for clinical details on the PCa patients, see Table 2).

sCMG2 was measurable in the serum of 335 subjects (84.4%), with a median concentration of 1.60 ng/mL (mean: 3.49 ng/mL; 0.2–27.01 ng/mL, median: 1.60 ng/mL). The median sCMG2 concentrations in the serum of the control subjects and the patients with a localized PCa are not significantly different (1.06 ng/mL vs. 1.02 ng/mL, *p* = 0.95, Mann–Whitney U-test). However, the mean sCMG2 concentration of patients with metastasized PCa is significantly increased (1.06 ng/mL vs. 1.73 ng/mL, *p* = 0.009, Mann–Whitney U-Test; see Figure 1).

The control subjects included a subcohort of men (*n* = 30) previously diagnosed and treated with PCa, who are tumor-free for five years. The probands in the tumor-free group did not show a significant different median sCMG2 concentration to either the other control subjects (1.53 ng/mL vs. 0.97 ng/mL) nor to the patients with PCa (1.53 ng/mL vs. 1.27 ng/mL; data not shown). 

Additionally, when comparing the sCMG2 concentration of patients with other tumor entities, significant differences can be observed to the median sCMG2 concentration in the serum of the control subjects (2.01 ng/mL vs. 1.06 ng/mL; *p* = 0.001; Mann–Whitney U-Test) and to the median sCMG2 concentration in the serum of patients with localized PCa (2.01 ng/mL vs. 1.02 ng/mL; *p* = 0.01; Mann–Whitney U-Test), but not to the median sCMG2 concentration in the serum of patients with metastasized PCa (2.01 ng/mL vs. 1.73 ng/mL; *p* = 0.68; Mann–Whitney U-Test; see Figure 1). Then, comparing the sCMG2 serum concentration of the different tumor entities subgroups comprised in the group “other tumor entities” (renal carcinoma, urothelial carcinoma, and others, see Table 1), no significant differences were observed (*p* = 0.66; Kruskal–Wallis test, data not shown).

### 2.2. sCMG2 Concentration and PCa Clinical Parameters

In bivariate correlation analyses, there was no significant association between the sCMG2 concentration and the prostate-specific antigen (PSA) level in PCa patients (*p* = 0.32; Spearman–Rho; *n* = 173) or in the whole study group (*p* = 0.28; Spearman–Rho; *n* = 377). However, when analyzing the different subgroups, a significant association between a lower sCMG2 concentration and an increased serum PSA could be detected in patients with localized PCa (*p* = 0.015, r_S_ = −0.23; Spearman–Rho), while in patients with metastasized PCa, there was no significant association between these parameters (*p* = 0.21; r_S_ = −0.16; Spearman–Rho). Also, sCMG2 concentration exhibited no significant correlation to age (*p* = 0.38; Spearman–Rho; *n* = 397). 

Then, analyzing the sCMG2 concentration in PCa patients according to the Gleason scores, we did not observe any significant differences between score 5 to 10 (*p* = 0.98; Kruskal–Wallis test; *n* = 157, see Figure 2a). Moreover, than analyzing three groups of combined Gleason scores (<7, *n* = 18; 7, *n* = 80; and >7, *n* = 59), no significant differences between the sCMG2 concentration among the three groups could be detected (*p* = 0.95; Kruskal–Wallis test, data not shown).

Then, analyzing the sCMG2 concentration in PCa patients according to the T stage, we detected a significant increase in T4 (*p* = 0.04; Kruskal–Wallis test). While median sCMG2 concentration in T1 to T3 was relatively similar (0.61–1.04 ng/mL; *n* = 107), in T4, the median sCMG2 concentration in the serum of the patients was nearly double as high (2.13 ng/mL; *n* = 43; see Figure 2b).

### 2.3. Diagnostic Capability of sCMG2

We further analyzed the usage of sCMG2 concentration in the serum for the discrimination of subjects with or without PCa, as well as for the discrimination of PCa patients with or without metastasized tumors. In receiver operating characteristic ROC analyses, there was no significant discriminative power for sCMG2 for subjects with or without PCa (*p* = 0.21; area under the curve AUC = 0.54; see Figure 3a). On the other hand, sCMG2 was observed to be a significant discriminative factor for patients with localized or metastasized PCa (*p* = 0.028). However, with an AUC of 0.6, sCMG2 remains a poor discriminator (see Figure 3b). 

### 2.4. Prognostic Value of sCMG2 in the Serum of PCa Patients

Lastly, we analyzed the association of sCMG2 concentration with the overall and disease-specific survival of the PCa patients cohort studied. Of 90 PCa patients with survival data, 22 died during the observation interval. However, owing to the increased age of the patients and sometimes existing comorbidities, only 13 cases of disease-specific survival were recorded. In Kaplan–Meier survival analyses, we observed a slightly worsened survival for PCa patients with higher (≥1.3 ng/mL) sCMG2 serum concentration. This effect was seen in the overall (see Figure 4a) and disease-specific (see Figure 4b) survival. While PCa patients with low sCMG2 concentration had an average of 165 months in overall survival, PCa patients with elevated sCMG2 concentrations had 112 months of average overall survival.

However, the impact of the sCMG2 serum concentration on the overall or disease-specific survival of the PCa patients was not significant (*p* = 0.29 and *p* = 0.21, respectively).

## 3. Discussion

In the present study, we measured the concentration of sCMG2 in the serum of PCa patients and control subjects and correlated the data to demographic, clinical, and prognostic parameters. To the best of our knowledge, this is the first report analyzing sCMG2 concentration in tumor patients for diagnostic purposes. We observed measurable sCMG2 concentrations in almost 85% of the analyzed samples, with a median concentration of 1.60 ng/mL.

Although there are some mechanistic studies on the cellular role of CMG2, studies about the pathophysiological action of sCMG2 are scarce. CMG2 was demonstrated to induce endothelial proliferation in human umbilical vein endothelial cells (HUVECs) and tumor vasculature [20], and thus was long discussed as an option for anti-angiogenic tumor therapy [28,29,30,31]. However, there are several reports that CMG2 exerts a pathophysiological role in the tumor cell itself. Ye and colleagues observed an overexpression of CMG2 mRNA in prostate carcinoma cell lines in comparison with prostate epithelial cell lines, and that the knockdown of CMG2 enhanced the invasiveness of prostate tumor cells [32]. In a recent study, CMG2 was identified to be capable of co-activating a Wnt/b-catenin pathway via LRP6 binding in gastric cancer cells, thus triggering the maintenance of a stem cell-like phenotype associated with the capability of epithelial-mesenchymal transition (EMT) and high metastatic abilities [33]. Most recently, in glioma cells, a CMG2-induced YES-associated protein (YAP) expression, associated with increased invasiveness, was observed [34]. In the present study, we also observed an association between elevated sCMG2 levels in the serum of the prostate carcinoma patients and an advanced tumor progression, characterized by a high T stage and metastasization. However, the question of whether this association is the result of aggressiveness-promoting properties of sCMG2, or whether increased sCMG2 concentrations are a surrogate marker of an advanced tumor stage, should be addressed in further studies.

CMG2 has been studied in a variety of tumor entities for diagnostic and prognostic reasons; however, mainly on the mRNA and cellular protein level. Ye and colleagues demonstrated an association between a decreased CMG2 mRNA expression and a higher tumor stage in breast cancer; additionally, decreased CMG2 mRNA levels were significantly associated with a worsened disease-specific and overall survival [35]. Furthermore, decreased CMG2 mRNA and cellular protein levels were also associated with a worsened disease-specific survival in soft tissue sarcoma patients [36]. However, Tan and colleagues recently observed that higher CMG2 mRNA and protein levels are associated with a poorer prognosis for glioma patients, thus imposing detrimental data to the former studies [37]. Furthermore, it was also demonstrated that in gastric cancer, an increased CMG2 expression was associated with advanced tumors, and that an elevated CMG2 mRNA and protein expression was an independent predictor for a poor prognosis [33]. Concordantly with the latter results, we found an increased sCMG2 concentration to be associated with a slightly worse PCa patient survival; however, this effect was not significant. Whether these contradictory results are because of the different tumor entities studied or patient cohorts used, or if there is a regulatory pathway controlling CMG2 expression in a multi-faceted way, remains to be elucidated.

Although a limitation of our study is that blood samples were not necessarily drawn at the initial diagnosis of PCa and that no repeated sampling during the progression of the disease could be performed, the observed correlations point towards a role of sCMG2 during PCa progression. Therefore, it remains to be elucidated whether sCMG2 serum concentrations at the initial PCa diagnosis have a predictive value for the course of the disease. Given this prerequisite, sCMG2 may be an interesting additional candidate for the early decision on a personalized PCa treatment regime (active surveillance versus surgery with/without radio- or chemotherapy). 

## 4. Materials and Methods 

### 4.1. Study Subjects and Blood Sampling

A total of 397 subjects were recruited for this study at the Clinic of Urology, University Hospital of the Martin Luther University Halle-Wittenberg. The study project was approved by the local ethics committee of the medical faculty of the Martin Luther University Halle-Wittenberg (Approval date: 06-07-2011). The patients gave written informed consent. The recruitment and recording of patients’ information were conducted in accordance with the declaration of Helsinki (2013). Demographic and clinical data were collected and are compiled in Table 1.

From every participant, 10 mL venous blood was drawn. Blood components were separated by centrifugation at 4000× *g* for 10 min. Serum and the cellular phase were stored individually at −80 °C upon usage.

### 4.2. ELISA Measurements

Serum samples were thawed on the day of ELISA measurement and centrifuged at 4000× *g* for 10 min. Then, 100 µL of serum supernatant was added per well on the CMG2 ELISA kit (antibodies-online, Aachen, Germany; ABIN420827; primary antibody: anti-human CMG2 polyclonal goat antibody). Hybridization and washing steps were performed according to the manufacturer’s protocol. Ten minutes after addition of the TMB substrate, absorption was measured at 450 nm on a microplate reader (GENios, Tecan, Männedorf, Switzerland). Standard curves with a standard quantitation range from 0.156 to 20 ng/mL were calculated for every plate separately, and the sCMG2 concentration in ng/mL for every sample was calculated from the according standard curve.

### 4.3. Statistical Analyses

All statistical analyses were performed in SPSS 25 (IBM, Armonk, NY, USA). Non-parametric tests (Mann–Whitney U-Test or Kruskal–Wallis test) were applied for the comparison of sCMG2 serum concentration between different groups. Bivariate correlation analyses according to Spearman–Rho were used for the testing of correlation of metric variables (sCMG2 concentration, PSA concentration, age). ROC analyses and survival analyses (Kaplan–Meier and Cox’s regression analyses) were applied to test the diagnostic and prognostic capabilities of the sCMG2 concentration for PCa. *p*-value < 0.05 was considered as significant.

## 5. Conclusions

sCMG2 concentration in the serum was measurable in a vast majority of the subjects, with significantly higher concentrations in patients with metastasized PCa and other tumors than in the control subjects or patients with localized PCa. sCMG2 concentrations were not significantly associated to Gleason score or age of the PCa patients, but serum PSA exhibited a significant association to sCMG2 concentration in patients with localized PCa. Furthermore, we observed a significant association between an increased sCMG2 concentration and the highest T stage. Elevated sCMG2 serum levels tend to be associated with a worsened survival of the PCa patients. Altogether, sCMG2 may be an interesting candidate for a complementary biomarker in the clinical management of PCa patients.

## Figures and Tables

**Figure 1 cancers-11-01059-f001:**
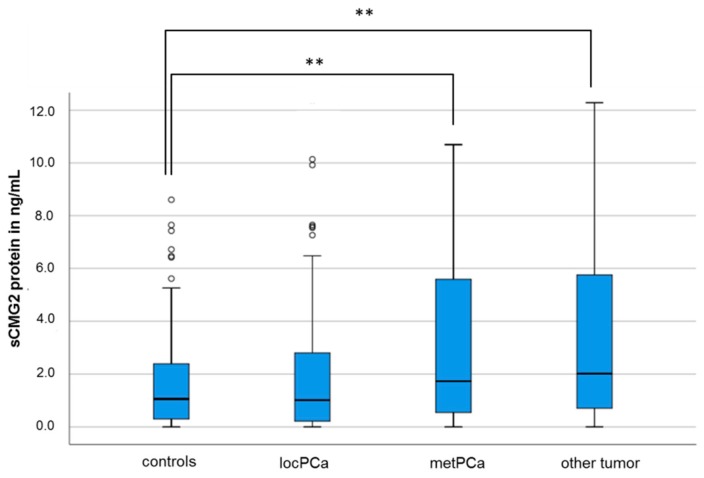
Soluble capillary morphogenesis gene 2 (sCMG2) concentration in the serum of control subjects and PCa patients. Comparison of sCMG2 concentration in control subjects and patients with localized or metastasized PCa and patients with other tumors. Abbreviations: locPCa, localized prostate carcinoma; metPCa, metastasized prostate carcinoma; ***p* < 0.01, Mann–Whitney U-Test.

**Figure 2 cancers-11-01059-f002:**
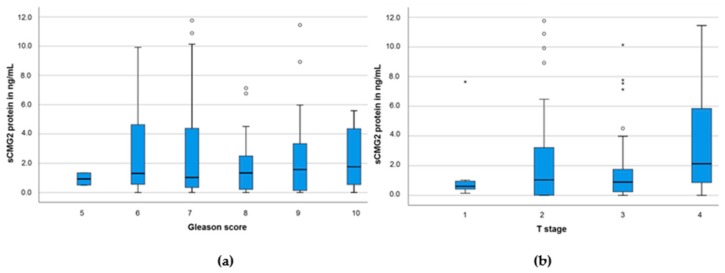
sCMG2 serum concentration in PCa patients with (**a**) different Gleason scores or (**b**) different T stages.

**Figure 3 cancers-11-01059-f003:**
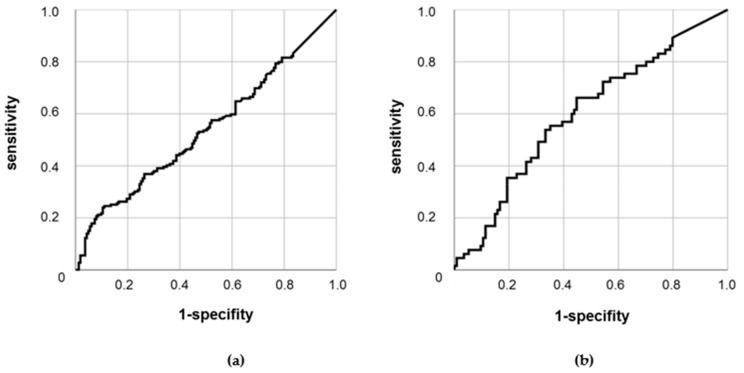
ROC analyses of sCMG2 serum concentration (**a**) in PCa patients (*n* = 179) vs. control subjects (*n* = 163); (**b**) in localized (*n* = 114) vs. metastasized (*n* = 65) PCa patients.

**Figure 4 cancers-11-01059-f004:**
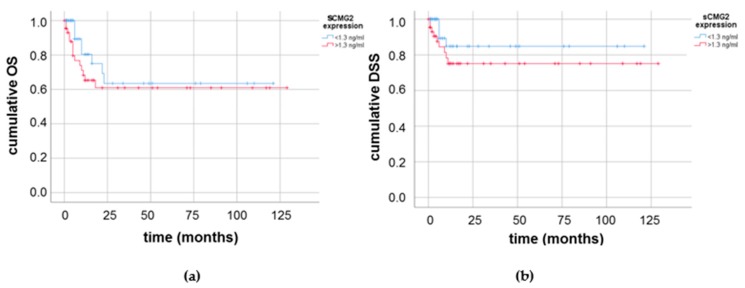
Kaplan–Meier survival analyses (**a**) for the overall survival (OS); or (**b**) the disease-specific survival (DSS) of PCa patients with elevated (≥1.3 ng/mL) in comparison with lower (<1.3 ng/mL) sCMG2 serum concentration.

**Table 1 cancers-11-01059-t001:** Distribution of age and prostate-specific antigen (PSA) concentration in the study cohort. PCa, prostate carcinoma.

	Diagnosis	*n*	Age (95% CI) ^1^	PSA (95% CI) ^2^
Control	BPH	83	66.5 (64.3–68.7)	3.4 (2.6–4.2)
subjects	Kidney-/bladder-related	41	61.4 (57.1–65.7)	1.9 (1.0–2.8)
	Testis-related	9	54.2 (47.6–60.8)	1.9 (0.9–3.0)
	PCa-free	30	67.5 (65.2–69.8)	0.16 (0.03–0.28)
PCa	Localized PCa	114	68.5 (67.2–69.8)	6.6 (4.9–8.3)
Patient	Metastasized PCa	65	70.2 (68.1–72.4)	186.5 (88.2–284.8)
Other	Renal carcinoma	25	72.2 (68.6–75.7)	2.2 (1.2–3.3)
tumor	Urothelial carcinoma	13	75.1 (69.5–76.7)	9.4 (1.0–26.8)
	other	17	69.9 (64.7–75.1)	8.2 (4.3–20.7)

^1^ mean age in years. ^2^ mean PSA concentration in ng/mL. Abbreviations: CI, confidence interval; PSA, prostate-specific antigen; BPH, benign prostatic hyperplasia.

**Table 2 cancers-11-01059-t002:** Clinical and prognostic parameters of the prostate carcinoma (PCa) patients.

		Localized PCa (*n* = 114)	Metastasized PCa (*n* = 65)
	5	2	0
	6	16	0
Gleason	7	66	14
score	8	10	11
	9	13	19
	10	0	6
	n.d.	7	15
	1	8	0
T	2	51	4
stage	3	33	11
	4	6	37
	n.d.	16	13
	No therapy/active surveillance	60	10
	RP	54	55
Therapy	Radiotherapy	24	32
	Androgen deprivation	23	53
	Chemotherapy	0	26
	No follow-up	69	20
status	Alive	43	25
	deceased	2	20

Abbreviations: n.d., not determined; PCa, prostate carcinoma; RP, radical prostatectomy.

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
