# Peer review of "Increased Soluble CMG2 Serum Protein Concentration Is Associated with the Progression of Prostate Carcinoma"

_cancers, 2019, doi:10.3390/cancers11081059_

Round 1

Reviewer 1 Report

Dr. Behre and team determine by liquid biopsy the serum CMG2 levels in PCa patients using age matched controls and correlate sCMG2 data to clinical and prognostic parameters. The manuscript proposes an interesting concept and applies a appropriate statistical tests. However, I was disappointed the Kruskal-Wallis test results in Fig 2a. Can the combination of PSA and sCMG2 distinguish <7 from >7 Gleason score? Also, is there a relationship between bone alkaline phophastase levels and sCMG2? Please add to section 4.2 the standard curve quantitation range for ELISA plates.

Author Response

Dr. Behre and team determine by liquid biopsy the serum CMG2 levels in PCa patients using age matched controls and correlate sCMG2 data to clinical and prognostic parameters. The manuscript proposes an interesting concept and applies a appropriate statistical tests.

(1) However, I was disappointed the Kruskal-Wallis test results in Fig 2a. Can the combination of PSA and sCMG2 distinguish <7 from >7 Gleason score?

We re-evaluated these variables according to the reviewer’s request. PSA levels were, as expected, significantly associated with the Gleason score (p = 0.024, Kruskal-Wallis test). A combination of sCMG2 and PSA levels also exhibited a significant correlation to the Gleason score (p = 0.031, Kruskal-Wallis test); however, it is likely that these correlation is mainly driven by the PSA level. When grouping sCMG2 and PSA levels according to the median (PSA: 6.1 ng/ml; sCMG2: 1.28 ng/µl) into three groups (both proteins low, n = 35; one protein elevated, n = 87; both proteins elevated, n = 31), we detected a trend towards significance when correlating to Gleason <7, 7 and >7 (p = 0.068; Pearson’s ChiSquare test). Analogously, we expect these correlations to be mainly driven by PSA, not sCMG2. Therefore, according to our results, we do not consider sCMG2 serum concentration as predictor for Gleason score of the PCa.

(2) Also, is there a relationship between bone alkaline phophastase levels and sCMG2?

We thank the reviewer for this helpful suggestion for further correlation analyses and the putative identification of sCMG2-related variables. Unfortunately, we do not have access to an applicable number of alkaline phosphatase levels in this PCa patient cohorts, therefore correlations between these two parameters were not possible in this study.

(3) Please add to section 4.2 the standard curve quantitation range for ELISA plates.

We modified the following sentence in section 4.2 to add this information: “Standard curves with a standard quantitation range from 0.156 – 20 ng/ml were calculated for every plate separately, and the sCMG2 concentration in ng/ml for every sample was calculated from the according standard curve.” (marked in yellow). Thanks to the reviewer’s remark, we also detected the usage of a misleading unit (ng/µl instead of ng/ml) throughout the text and corrected this mistake (also marked in yellow).

Finally, we want to thank the reviewer for the helpful comments and remarks.

Best regards.

Thomas Greither (in behalf of the authors)

Reviewer 2 Report

The manuscript titled "Increased soluble CMG2 serum protein concentration 2 is associated with the progression of prostate 3 carcinoma" is an interesting study. Authors ave investiated role of CMG2 as biomarker for advance stage or metastatic stage biomarker. This study is quite relevant in prostate cancer progression prediction.However, there are  number of concerns with the manuscript in current form.

Authors should show the level of sCMG2 in patients with temporal progression of metastatic PCa.

Authors should compare level of sCMG2 with PSA  in localized and metastatic PCa.

Authors should clearly discuss for which decisions in the clinic they see additional value of sCMG2 in discussion section.

Authors should also comment about level of sCMG2 and months or years survival of the patients.

Author Response

The manuscript titled "Increased soluble CMG2 serum protein concentration 2 is associated with the progression of prostate 3 carcinoma" is an interesting study. Authors ave investiated role of CMG2 as biomarker for advance stage or metastatic stage biomarker. This study is quite relevant in prostate cancer progression prediction. However, there are  number of concerns with the manuscript in current form.

(1) Authors should show the level of sCMG2 in patients with temporal progression of metastatic PCa.

According to the reviewer’s remark, we analyzed the correlation of the time (in months) between the blood sampling and the end of follow-up via bivariate correlation analyses according to Spearman-Rho in patients with metastatic PCa.  There was no correlation between the observation time and the sCMG2 level (rS = 0.01; p = 0.948; see also Figure attached). However, it has to be stated, that this group comprises patients deceased from the PCa, patients deceased due to other circumstances and patients who were still alive at the end of the follow-up period.

Additionally, we unfortunately do not have access to a series of blood samples from the same patient during the progression of the metPCa. Therefore, this study remains a cross-sectional study, with blood samples not necessarily drawn at the date of metPCa diagnosis. However, we want to thank the reviewer for this hint for further studies.

(2) Authors should compare level of sCMG2 with PSA  in localized and metastatic PCa.

We thank the reviewer for this valuable remark. We compared sCMG2 and PSA levels in patients with localized or metastatic PCa via bivariate correlation analyses according to Spearman-Rho. While in patients with metastasized PCa, sCMG2 levels were not significantly associated to PSA levels (rS = -0.16; p = 0.21), in patients with localized PCa a mild, yet significant, inverse correlation could be observed (rS = -0.23; p = 0.015). We added this sentence to the manuscript (l.98 – 102):  However, when analyzing the different subgroups, a significant association between a lower sCMG2 concentration and an increased serum PSA could be detected in patients with localized PCa (p = 0.015, rS = -0.23; Spearman-Rho), while in patients with metastasized PCa there was no significant association between these parameters (p = 0.21; rS = -0.16; Spearman-Rho). (marked in yellow).

(3) Authors should clearly discuss for which decisions in the clinic they see additional value of sCMG2 in discussion section.

According to the reviewers suggestion, we added the following paragraph to the discussion section (l.178-184): Although being a limitation of our study that blood samples were not necessarily drawn at the initial diagnosis of PCa and that no repeated sampling during the progression of the disease could be performed, the observed correlations point towards a role of sCMG2 during PCa progression. Therefore, it remains to be elucidated whether sCMG2 serum concentrations at the initial PCa diagnosis have a predictive value for the course of the disease. Given this prerequisite, sCMG2 may be an interesting additional candidate for the early decision on a personalized PCa treatment regime (active surveillance versus surgery and/or radio- or chemotherapy).” (marked in yellow).

(4) Authors should also comment about level of sCMG2 and months or years survival of the patients.

We observed differences in survival time of 165 months in PCa patients with low sCMG2 (<1.3 ng/ml) to 112 months in PCa patients with elevated sCMG2 serum levels (≥1.3 ng/ml). As stated in the discussion section (l. 174-175), we consider this as a slight differences, although it was not significant. However, as the observation is in line with the recent publication of Tan et al., 2018, regarding the prognostic value of CMG2 protein in glioma tissue, we decided to include this information in the present manuscript.

Finally, we want to thank the reviewer for the helpful comments and remarks.

Best regards.

Thomas Greither (in behalf of the authors)
